# Assessment of Wetland Restoration and Climate Change Impacts on Water Balance Components of the Heeia Coastal Wetland in Hawaii

**Kariem A. Ghazal [1],\*, Olkeba Tolessa Leta [2] , Aly I. El-Kadi [3,4] and Henrietta Dulai [4]**

[1] Department of Soil Sciences and Water Resources, University of Kufa, El- Najaf 54003, Iraq

[2] Bureau of Watershed Management and Modeling, St. Johns River Water Management District, Palatka, FL 32177, USA; OLeta@sjrwmd.com

[3] Water Resources Research Center, University of Hawaii at Manoa, Honolulu, HI 96822, USA; elkadi@hawaii.edu

[4] Department of Earth Sciences, University of Hawaii at Manoa, Honolulu, HI 96822, USA; hdulaiov@hawaii.edu

\* Correspondence: kareem.alzubaidi@uokufa.edu.iq; Tel.: +96-477-3758-4068

**Abstract:** Hydrological modeling is an important tool that can be used to assess water resources' availability and sustainability that are necessary for food security and ecological health of coastal regions. In this study, we assessed the impacts of land use and climate changes on water balance components (WBCs) of the Heeia coastal wetland. We developed a Soil and Water Assessment Tool (SWAT) model to capture the unique characteristics of the Hawaiian Islands, including its volcanic soil's nature and high initial infiltration rates. We used the sequential uncertainty fitting algorithm to assess the sensitivity and uncertainty of WBCs under different climate change scenarios. Results of the statistical analysis of daily streamflow simulations showed that the model performance was within the generally acceptable criteria. Under future climate scenarios, rainfall change was the determinant factor most negatively impacting WBCs. Recharge and baseflow components had the highest sensitivity to the combined effects of land use and climate changes, especially during dry season. The uncertainty analysis indicated that the streamflow is projected to slightly increase by the middle of 21st century, but expected to decline by 40% during the late 21st century of Representative Concentration Pathways (RCP) 8.5.

**Keywords:** Heeia; Hawaii; climate change; SWAT model; water balance; wetland restoration

## 1. Introduction

Land-use change (LUC) and climate change (CC) are considered the main determinant factors for the changes of water balance components, nutrient fluxes, and thermal energy of a watershed. These factors are also expected to affect each other through the interaction of various physical, chemical, and biological processes [1,2]. For example, in the Hawaiian Islands, fossil evidence indicated that the combined interactions among CC, LUC, and biological invasions substantially aggravated the direct impacts of CC [3]. Examples of the human activities that aggravate the CC impacts include the conversion of wetlands into urban land, intensive agriculture practices, and the excessive use of fossil fuels. Consequences of these interactions are reflected in the implication of regional projects, such as plans for coastal wetland restoration or for sustainable development of water resources [4–8]. Hydrological modeling has a potential role in facilitating strategic decision-making concerning environmental response and in developing adaptation strategies to CC as well as polices for hazard

mitigation [7,8]. Actions can ensure optimized allocation of water resources under climate and land use changes [9,10].

The main consequences of climate change in tropical pacific ecosystems include an increase in temperature and evapotranspiration (ET), decrease in rainfall and runoff, and increase in sea level rise, which in turn affects the natural ecosystems [11,12]. For example, wetland ecosystems may undergo a decline in their functional capacity and shift in their geographic location [13]. Therefore, CC makes wetland restoration and management more complex due to its effect on ecological and hydrological interactions. Policymakers and restoration practitioners should take into account the potential impacts of CC during the implementation of wetland restoration projects. However, CC adaption and hazard mitigation strategies still depend on additional studies due to the influence of global CC on the local dynamics of freshwater availability [14].

Evidence of climate change impacts on the Hawaiian Islands has already been observed. Some of these impacts include changes in groundwater recharge and surface runoff, land sliding, soil erosion, and the degradation of coastal ecosystems, such as coral reefs [15,16]. Predictions include a decrease in prevailing northeasterly trade winds, rainfall amounts, groundwater recharge, and total streamflow [4,17,18]. Reduction in aquifer recharge negatively affects fresh water supplies. It is also expected that local impacts of climate change include warming of the air temperatures by over 0.17 °C per decade and an increase in ocean surface temperature by as much as 0.23 °C per decade [18]. Conditions can even worsen as the temperature trend is expected to increase by 1.3 to 2.7 °C by the end of the 21st century [18–20]. Ocean acidity is projected to increase by 30% and sea levels may rise between 1.5 to 3.3 cm per decade [18,21]. These impacts threaten human health by increasing pathogen abundances and by spreading invasive species [22]. The ocean may also undergo changes in circulation and in nutrient distribution, which may affect marine biota. Damages are expected to occur regarding infrastructures located in low lying areas due to sea level rise and beach loss [15,16,18].

Previous studies in Hawaii [6,17,22] have also indicated that the combined effect of increasing groundwater withdrawals and overall decrease in precipitation are expected to diminish groundwater baseflow, and total streamflow [22–24]. Regional climate change models predicted that the wet windward parts of the Hawaiian Islands might become wetter or remain stable in their seasonal rainfall, while the dry leeward sides would get drier [20]. Variation in solar radiation may occur due to variability in atmospheric transmissivity and cloud radiative properties [25]. Historical solar radiation anomalies showed a decreasing trend during the wet season and an increasing trend during the dry season, which directly affect ET [26]. The observed and projected changes of the Hawaiian ecosystems mandate that communities be better prepared for combating the effects of the global CC on water resources and ecosystem services. Models can take an important lead in this regard.

Watershed models, such as the Soil and Water Assessment Tool (SWAT) [27], coupled with regional climate models (RCM) have been used to assess the impacts of CC on the water balance components (WBCs) of watersheds [27,28]. There is a number of hydrological modeling and climate change studies on the leeward side of Oahu Island [19,21,22,29,30]. However, to date, there are very limited hydrological studies for the windward side of the island, especially for the Heeia watershed [4–6,31]. Therefore, integrated watershed model development and climate change assessment is urgently needed for the windward side of Oahu. In this study, we developed a SWAT model for the Heeia watershed and used the model to assess the impacts of CC and LUC on the WBCs in a wetland within the watershed. The model was designed to capture the unique characteristics of the Hawaiian Islands including their volcanic soils' nature and high initial infiltration rates. The main objective of the study was to develop an assessment methodology and provide results that are useful for decision makers towards efficient wetland management strategies.

## 2. Materials and Methods

### 2.1. Study Area

The Heeia wetland is the coastal part of the Heeia watershed, located on the windward side of the northeast coast of Oahu (Figure 1). The watershed is bounded by the Heeia fishpond and Kaneohe Bay to the east, and the crests of Koolau Mountains to the west. The Haiku and Iolekaa streams are the major perennial streams of the watershed [32,33]. Historically, the Heeia wetland was considered as one of the highest productive coastal areas on Oahu due to extensive taro and rice cultivation. In addition, the region is considered as an important economic resource due to the existence of the largest fishpond on Oahu located at the Heeia stream estuary.

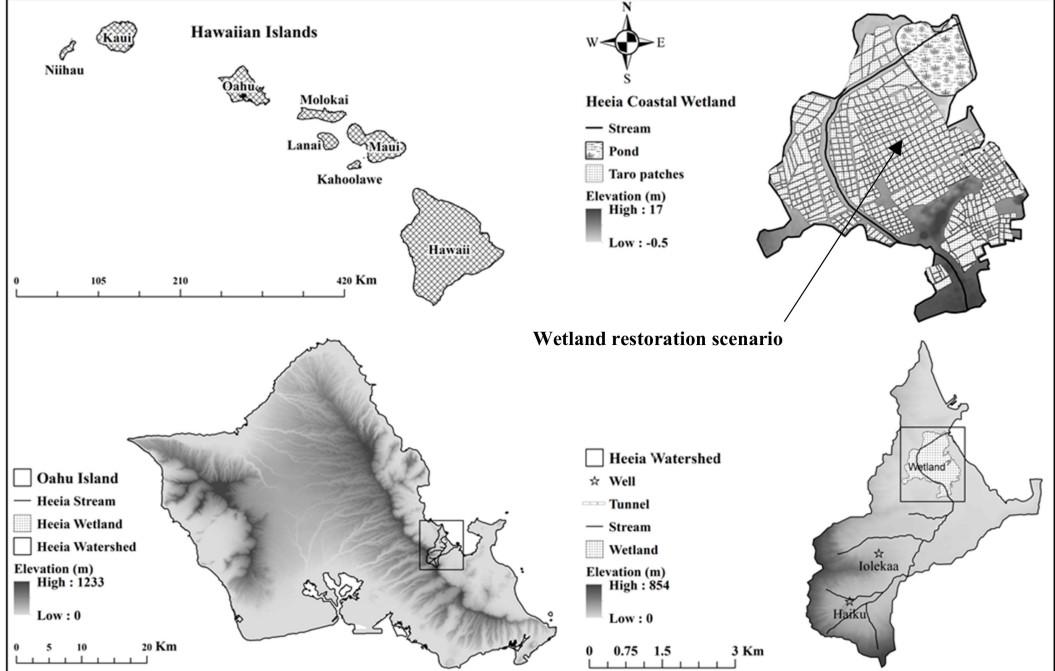

**Figure 1.** Geographic and topographic maps of the Heeia Coastal Wetland.

The Heeia watershed is a precipitous, rugged, and narrow valley that transforms into a flat swamp region near the shoreline, which represents about half of the coastal plain [34]. The watershed has an area of 11.5 km² with a total stream length of about 12 km. The Heeia coastal wetland represents the drainage basin of the main streams of the watershed and thus it is considered a major reservoir of freshwater. Before the 1950s, these hydrologic features enabled the indigenous population to meet their needs from land and ocean in a prized coastal region [35,36]. The elevation of the Heeia watershed ranges from 0 to 854 m above mean sea level (masl) with an average slope of 40% [6]. In comparison, the elevation of the Heeia wetland ranges from −0.5 to 17 masl with an average slope of 5% [32]. The wetland composition is dominated by emergent wetland (77%), forested wetland (8%), shrub wetland (5%), evergreen (4%), and other land use (6%) (Figure 2). The climate of Heeia is considered semi-tropical, which lies in the northeast trade-wind belt [37]. Rainfall is plentiful during the winter seasons from November through April, in contrast to the dry season, which extends from May to October [20]. The air temperature ranges between 20 °C to 28 °C throughout the year. Temperature is highest during August to September and lowest in January to February. The yearly average evapotranspiration rate in the Heeia watershed is about 840 mm [31,38].

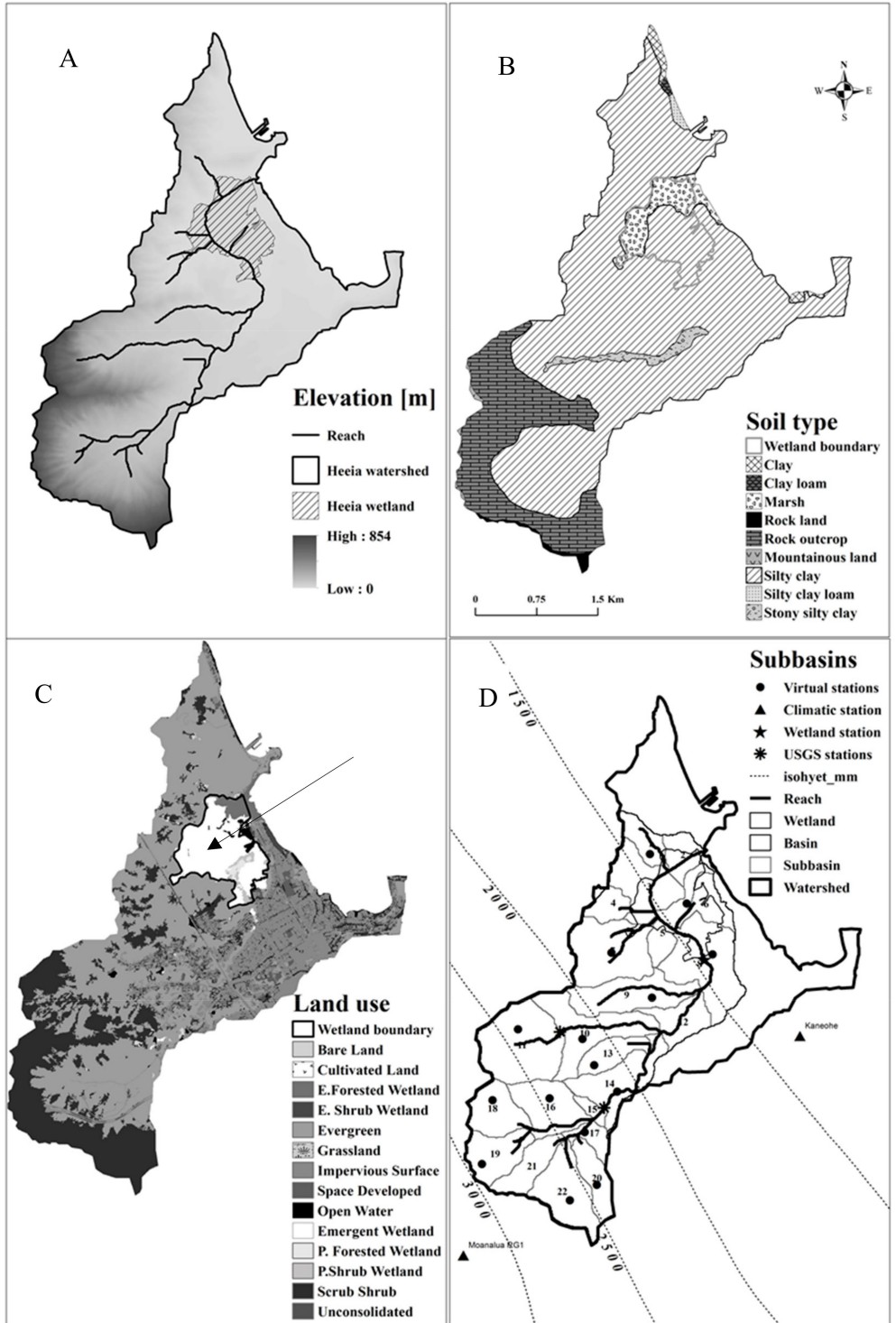

**Figure 2.** Digital Elevation Model (DEM) (**A**) of the Heeia watershed, soil type (**B**), land use (**C**), and delineated sub-watersheds with corresponding flow gauging locations and climatic stations (**D**).

## 2.2. Hydrological Modeling

A SWAT model [39,40] was developed for the Heeia watershed, based on available hydro-metrological and geo-spatial data. The model was run at a daily time-step for the period of 1/1/2000 to 12/31/2014. The model simulation time was split into three periods, namely, a warming-up period (2000–2001), a calibration period (2002–2008), and a validation period (2009–2014). The warming up period was used to initialize the state variables of the system. The modeled Heeia

watershed was divided into 22 subbasins and 984 hydrologic response units (HRUs) based on the similar combinations of land use, soil type, and slope. The daily streamflow data at the United States Geological Survey (USGS) station at Haiku and Wetland station were used for model sensitivity analysis (SA), calibration, and validation. Prior to calibration, a SA was performed using the Latin Hypercube-One-factor-At-a-Time (LH-OAT) technique of the SWAT Calibration and Uncertainty Program (SWAT-CUP) [41]. After auto-calibration using the Sequential Uncertainty Fitting (SUFI2) algorithm of SWAT-CUP, a manual calibration was then carried-out to fine tune the auto-calibrated parameter values, particularly to obtain a reasonable agreement for WBCs [42]. Such an approach substantially reduced the time-consuming manual calibration and also allowed quantitative and qualitative comparisons to be made [39].

*2.3. Climate Change Scenarios*

The SWAT model streamflow was used to evaluate the separate and combined effects of LUC and CC on the WBCs at the wetland and the Heeia watershed scales. The model was used to simulate the impact of the CC scenarios on streamflow by manipulating the climatic input data, which includes rainfall, temperature, and solar radiation.

For rainfall change scenarios, the statistically downscaled (250 × 250 m) seasonal rainfall anomalies [20] were used. Timm et al. [20] reported the values over the representative 30 years for the middle (2041–2070) and late (2071–2100) 21st century under Representative Concentration Pathways (RCP) 4.5 and 8.5 scenarios. Hereafter, the two periods are called 2050s and 2080s, respectively. The RCP 4.5 scenario refers to the radiative forcing of 4.5 w/m$^2$ with greenhouse gas concentration of 650 ppm, and temperature anomaly of 2.4 °C. The respective values for the RCP 8.5 are 8.5 w/m$^2$, 1370 ppm, and 2.9 °C. The projected rainfall anomalies for Hawaii were spatially interpolated and made available as Geographic Information System (GIS) layers by Timm et al [20] (see Figure 3 as an example). The spatially interpolated data included eight GIS map coverages that represented two dry seasons (2050s, 2080s) and two wet- seasons (2050s, 2080s) per each scenario. The SWAT model requires rainfall values in the form of one rain gauge data per each subbasin. Hence, the spatially interpolated statistical rainfall anomalies could not be directly used by the model. To capture the lower (minimum rainfall) and upper (maximum rainfall) anomalies per subbasin, observed daily rainfall values were perturbed for each subbasin based on the estimated lower and upper rainfall anomalies (Figure 3). This approach resulted in two additional scenarios, where the first one was for the lower limit and the other for the upper limit for each of the RCP 4.5 and 8.5 scenarios. In this study, the lower and upper bound rainfall are labeled as rainfall min and rainfall max, respectively. Such an approach was also used in other studies [22]. The baseline rainfall values were increased or decreased through multiplication by the factors obtained from the projected rainfall anomalies. The perturbation values (or percent changes) were implemented in SWAT's subbasin input files [39] to reflect future seasonal rainfall changes. Daily climatic data of the watershed for the period 2000 to 2014 were used as a baseline.

For temperature and solar-radiation change scenarios, historical data were perturbed based on other studies that reported expected change values for the Hawaiian Islands [19,43]. For the RCP 4.5 scenario, the temperature was increased by 1 °C for the 2050s and 1.5 °C for the 2080s. The respective increases for the RCP 8.5 scenario were 1.5 °C and 2 °C. Solar radiation was increased by 5% (2050s) and 10% (2080s) for the RCP 4.5 scenario, and by 10% (2050s) to 15% (2080s) for the RCP 8.5 scenario. Wet season solar radiation data were decreased by the same magnitudes. These absolute changes for temperature and percent changes for solar radiation were implemented in SWAT input subbasin files to perturb historical temperature and solar radiation values of 2000 to 2014.

The estimated output data included the lowest and highest impacts of combined climatic variables (rainfall, temperature, and solar radiation) on the WBCs at the Wetland and the whole Heeia watershed scales. The combined effects of LU and CC scenarios of RCP 4.5 and 8.5 were simulated and formulated into eight combinations.

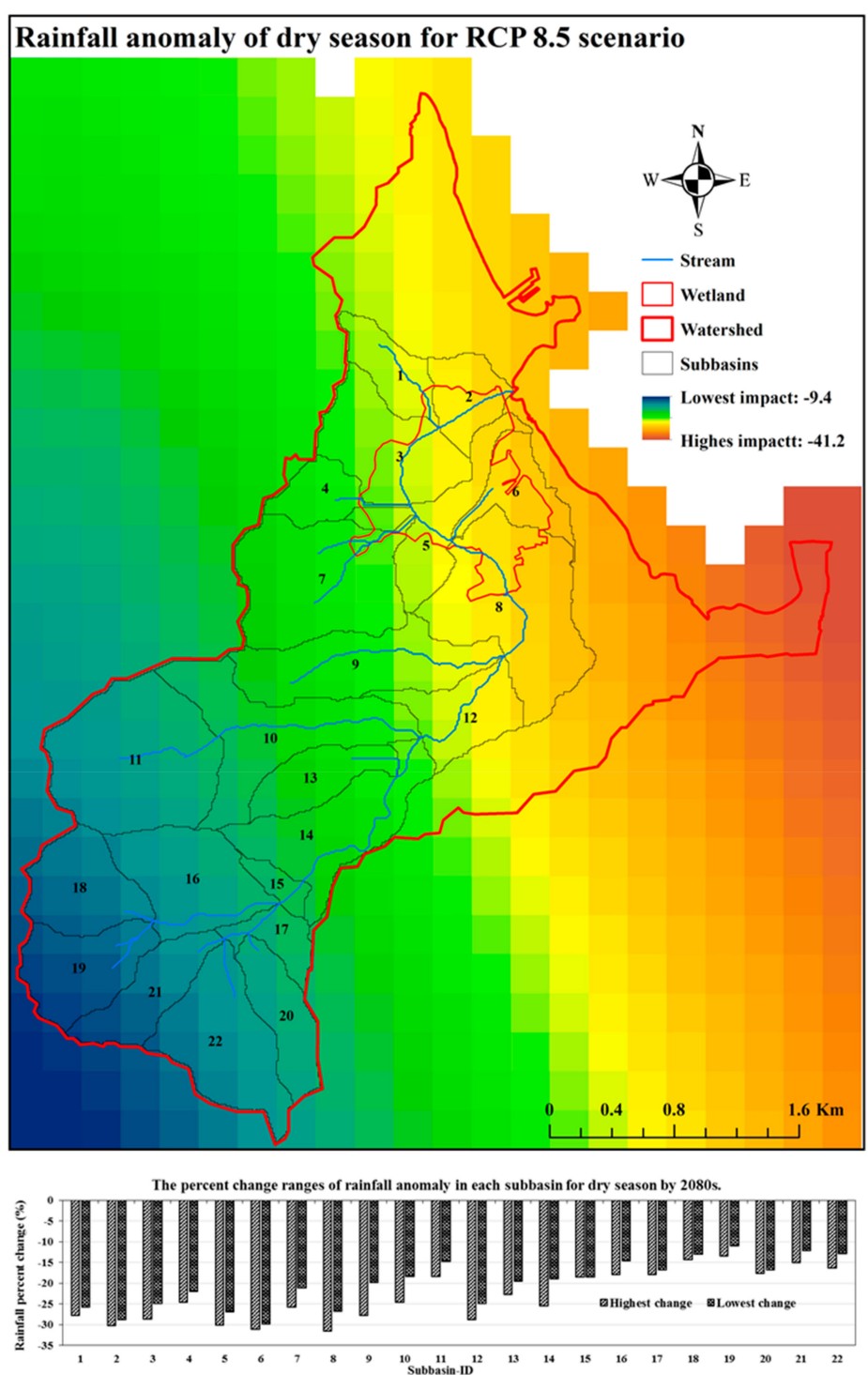

**Figure 3.** Projected rainfall anomaly adapted from (Timm et al. [22]) overlaid with the delineated subbasins (**top**) and the minimum and maximum rainfall change values within the corresponding subbasins (**bottom**).

## 2.4. Land Use Change Scenario

The LUC scenario concerned the conversion of current coastal wetland land uses based on the planned coastal wetland restoration of the watershed [44]. The plan includes the conversion of the current dominant invasive species California grass (*Brachiara mutica*) to organic wetland taro (*Colocasia esculenta*). In addition, the plan calls for converting the existing wetland mangrove forest, also an invasive species, into a pond serving as a native habitat for aquatic species. Based on land use maps,

perennial California grassland mainly exists in the coastal wetland (Figure 2), approximately covering 7% of the modeled area of 8.5 km$^2$. In addition, eight hectares of wetland mangrove forest (1% of the modeled area) is located around the Heeia stream estuary. As of 2019, restoration efforts have progressed and a significant portion of the mangrove has been removed, which was not included in this study's baseline model. Consequently, the current estuarine forested wetland in the current land use map was treated as water code in the SWAT database during LUC treatment, while the existing California grassland was converted into taro cultivation. The taro crop was chosen in the restoration plan because it is an important staple food and spiritual plant in Hawaiian cultural heritage. Moreover, until 1940s, the Heeia wetland was actively cultivated with taro [44]. To reflect the land use conversion to taro field, the current land use database of SWAT was updated for this study by adding taro properties into the existing database of the model. The properties and variables that were changed in the database are summarized in Table 1. These parameters were adopted based on literature values [40,45] and field measurements. Furthermore, some variable values of herbaceous land use from the SWAT database were used for wetland taro, considering that this plant is classified as herbaceous perennial tropical crop [46]. In addition, since taro cultivation requires water ponding and irrigation, streamflow diversion was activated during LUC scenario implementation. The streamflow diversion for taro's irrigation-water requirement from streamflow reach was set at least to 50% of minimum streamflow to avoid stream channel drying. The effects of LUC on simulated WBCs were analyzed on seasonal basis at both the wetland and the watershed scales.

**Table 1.** Brief description of the variables in the Soil and Water Assessment Tool (SWAT) plant growth database file of wetland taro.

| Variable Name | Code and Values | Definition | Reference |
|---|---|---|---|
| ICNUM | 142 | Land cover/plant code | This study |
| CPNM | TARO | Four-character code of land name | This study |
| IDC | 6 | Herbaceous perennial crop code | [39,46,47] |
| CROPNAME | Wetland Taro | Name of flooded taro | This study |
| BIO_E | 47 | Radiation-use efficiency of herbaceous | [39,47] |
| HVSTI | 0.01 | Harvest index for optimal growth | [45] |
| BLAI | 2.5 | Maximum potential leaf area index (LAI) | [48] |
| FRGRW1 | 0.11 | Fraction of the plant growing season | [39] |
| LAIMX1 | 0.13 | Fraction of the maximum LAI (first point) | [45] |
| FRGRW2 | 0.24 | Fraction of the plant growing season | [45] |
| LAIMX2 | 0.91 | Fraction of the maximum LAI (second point) | [39,45] |
| DLAI | 0.89 | Fraction of growing season (decline leaf area) | [39,45] |
| CHTMX | 0.7 | Maximum canopy height (meter) | This study |
| RDMX | 0.6 | Maximum root depth (meter) | This study |
| T_OPT | 25 | Optimal temperature for plant growth (°C) | This study |
| T_BASE | 21 | Minimum temperature for plant growth (°C) | This study |

## 3. Results and Discussion

### 3.1. SA and Streamflow Simulation

SA identified curve number at moisture condition II (CN2), effective hydraulic conductivity in the main channel (CH_K2), baseflow factor (ALPHA_BF), channel Manning's roughness coefficient (CH_N2), lateral flow travel time (LAT_TIME), saturated soil hydraulic conductivity (SOL_K), minimum

depth for groundwater flow occurrence (GWQMN), and groundwater recharge to a deep aquifer (RCHRG_DP) as the most sensitive parameters as they showed lower *p* value and higher absolute *t*-statistics values (Table 2). The optimized values of sensitive parameters are listed in Table 3 while the model performance evaluation is summarized in Table 4. The optimized parameter values were physically acceptable considering the hydrological features of the Heeia watershed (Figure 2). The statistical evaluation results for daily streamflow simulation (Table 4) showed that the model performance was within the generally acceptable criteria for model evaluation, especially under scarcity of climate data. Overall, based on the recommended quantitative statistics (Nash-Sutcliffe Efficiency (NSE), Root Mean Squared Error (RMSE) to observation Standard deviation Ratio (RSR), and Percent Bias (PBIAS)), the model simulation could be judged as satisfactory because the averages of the three criteria were 0.53, 0.66, and 5.9 (Table 4), respectively, during the calibration and validation periods [49,50].

**Table 2.** SWAT parameter sensitivity to daily streamflow at the Haiku station. Acronyms are explained in Table 3.

| Parameter | t-Stat | *p*-Value | Parameter | t-Stat | *p*-Value |
|---|---|---|---|---|---|
| CN2 | −50.73 | 0 | SURLAG | 1.289 | 0.198 |
| CH_K2 | 34.071 | 0 | OV_N | −1.031 | 0.303 |
| ALPHA_BF | −16.563 | 0 | EPCO | 0.992 | 0.322 |
| CH_N2 | 6.242 | 0 | GW_DELAY | −0.686 | 0.493 |
| LAT_TTIME | 4.145 | 0 | SLSUBBSN | 0.677 | 0.499 |
| SOL_K | 2.69 | 0.007 | SLSOIL | 0.647 | 0.518 |
| GWQMN | 2.564 | 0.011 | HRU_SLP | 0.617 | 0.537 |
| RCHRG_DP | −1.805 | 0.72 | REVAPMN | 0.505 | 0.614 |
| ESCO | −1.672 | 0.095 | GW_REVAP | −0.327 | 0.744 |
| SOL_AWC | 1.496 | 0.135 | SOIL_Z | −0.299 | 0.765 |
| CANMX | 1.411 | 0.159 | | | |

**Table 3.** Optimized SWAT sensitive parameter values of the Heeia watershed at Haiku and Wetland stations.

| Parameter | Description | Unit | Range | | Calibrated | |
|---|---|---|---|---|---|---|
| | | | Minimum | Maximum | Haiku | Wetland |
| **ALPHA_BF** | Baseflow factor | day$^{-1}$ | 0 | 0.005 | $3 \times 10^{-4}$ | 0.0045 |
| **CANMX** | Maximum canopy storage | mm | −0.4 | 0.4 | 0.1 | −0.3 |
| **CH_K2** | Effective hydraulic conductivity in main channel | mm h$^{-1}$ | 10 | 50 | 39 | 20.4 |
| **CH_N2** | Channel Manning's roughness coefficient | | 0.02 | 0.07 | 0.02 | 0.04 |
| **CN2** | Curve number at moisture condition II | | −0.5 | 0.1 | −0.49 | −0.47 |
| **ESCO** | Soil evaporation compensation factor | | 0.5 | 1 | 0.9 | 0.5 |
| **LAT_TIME** | Lateral flow travel time | day$^{-1}$ | 10 | 90 | 81 | 18 |
| **RCHRG_DP** | Groundwater recharge to deep aquifer | | 0 | 0.05 | 0.045 | 0.0002 |
| **GWQMN** | Minimum depth for groundwater flow occurrence | mm | 1 | 1000 | 137 | 774.5 |
| **SOL_K** | Saturated soil hydraulic conductivity | mm h$^{-1}$ | −0.5 | 0.1 | −0.4 | −0.03 |
| **SOL_AWC** | Soil water available capacity | | −0.2 | 0.3 | −0.03 | 0.16 |
| **SURLAG** | Surface runoff lag coefficient | day$^{-1}$ | 0.5 | 2.5 | 1 | |

**Table 4.** The statistical evaluation summarizes the results of the Heeia watershed model at Haiku and Wetland stations (Figure 2).

| Station | Period | Time Span | NSE | PBIAS (%) | RSR | r | P-factor | R-factor |
|---|---|---|---|---|---|---|---|---|
| Haiku | Calibration | 2002–2008 | 0.60 | 4.60 | 0.66 | 0.69 | 0.96 | 1.36 |
|  | Validation | 2009–2014 | 0.51 | 8.00 | 0.70 | 0.54 | 0.96 | 0.89 |
| Wetland | Calibration | 2002–2008 | 0.51 | 13.00 | 0.63 | 0.67 | 0.81 | 0.81 |
|  | Validation | 2009–2014 | 0.50 | −2.59 | 0.67 | 0.50 | 0.95 | 0.67 |

The daily streamflow simulation results with the 95% model prediction uncertainty are shown in Figures 4 and 5, respectively. Results generally indicated an overall good performance of SWAT for the watershed as the model reproduced the temporal evolution of the daily hydrograph with acceptable statistical values (Table 4, Figures 4 and 5). As seen in the figures, more than 80% of the observed daily streamflows are bracketed within the 95% prediction uncertainty (Figures 4 and 5). Both streamflow hydrographs and statistical indexes demonstrated the applicability of the model for the watershed, which provided confidence in its use in future streamflow predictions. The next section presents the implications and impacts of CC and LUC on water balance components of the Heeia coastal wetland.

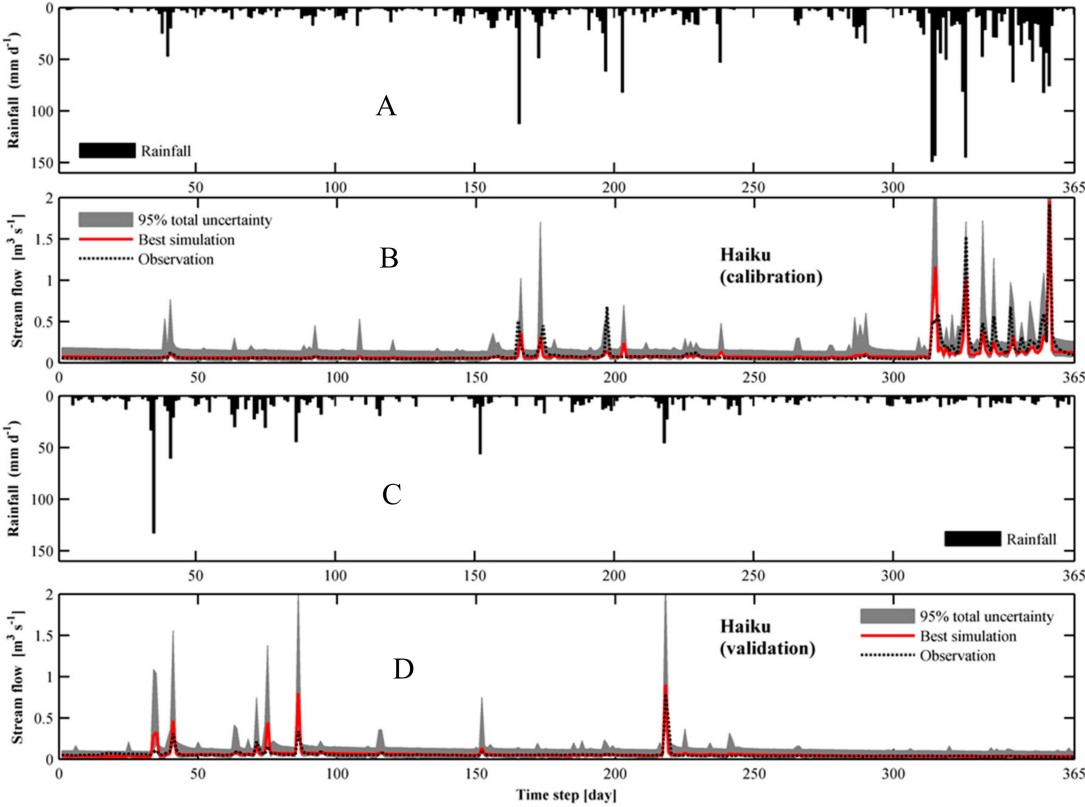

**Figure 4.** Areal average daily rainfall (panels **A** and **C**) and the respective simulated and observed streamflow with 95% prediction uncertainty (panel **B** and **D**) at the Haiku station for one year of the calibration period (2002–2008).

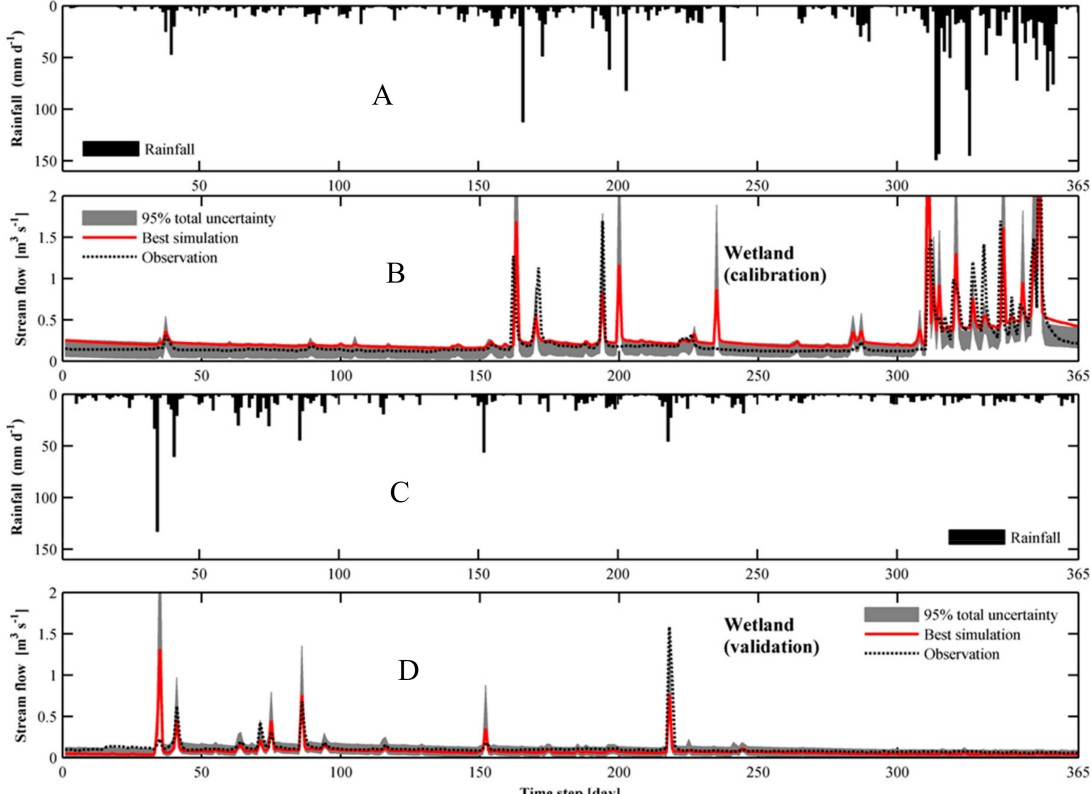

**Figure 5.** Areal average daily rainfall (panels **A** and **C**) and the respective simulated and observed streamflow with 95% prediction uncertainty (panel **B** and **D**) at the Wetland station for one year of the calibration period (2002–2008).

## 3.2. Impacts of LUC on WBCs

Conversion of the existing wetland's California grass to taro cultivation causes an overall decrease in total streamflow (see Figure 6 as an example). Such an effect should be expected due to the diversion of stream water for taro irrigation and to the increased evaporation from ponding water (Table 5). However, in general, the change in the WBCs at watershed scale was insignificant, which should be expected due to the small size of the restored area compared to that of the whole watershed. LUC results showed decreases in groundwater recharge and baseflow. However, increases in surface runoff and lateral flow were projected at both the wetland and watershed scales. Further, compared with the other WBCs, surface runoff and lateral flow showed the highest positive change. This could be mainly due to the nature of the taro cultivation that includes ponding of water and the implementation of an impervious layer at a depth of 25 cm below the soil surface [51]. Such factors should have a significant influence by reducing infiltration and increasing surface runoff during the wet season (Table 5 and Figure 6).

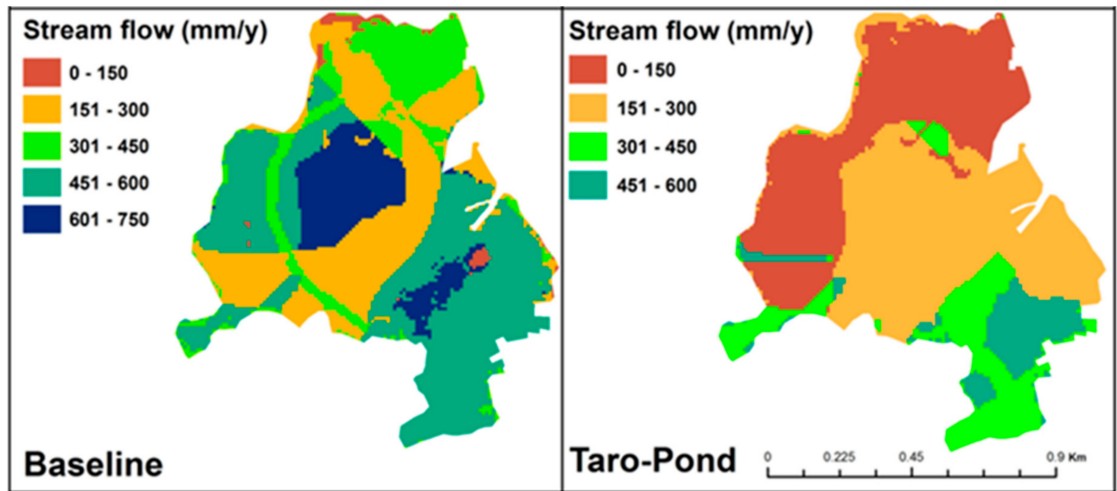

**Figure 6.** Yearly average streamflow within the Heeia wetland for baseline and land use change (LUC).

**Table 5.** Seasonally-averaged percent change in the water balance components (WBCs) of the Heeia wetland due to LUC relative to the baseline.

| Season | Rainfall | Streamflow | Runoff | Lateralflow | Baseflow | Recharge | Soil Moisture | ET | PET |
|---|---|---|---|---|---|---|---|---|---|
| wet | 0.00 | 19.22 | 80.95 | 40.78 | −42.07 | −41.42 | 24.01 | −4.29 | −0.27 |
| dry | 0.00 | −12.17 | 13.32 | 85.22 | −41.37 | −43.07 | 57.49 | 5.54 | 0.26 |

*3.3. Impacts of CC on WBCs*

In comparison with the LUC scenario, CC is expected to cause further decrease in baseflow and groundwater recharge (Table 6 and Figure 7). This could be due to a consistent decrease in rainfall for both wet and dry seasons. In addition, it is noticed that unlike the LUC, the surface runoff is projected to be seasonally dependent. For example, up to 44% decrease in surface runoff is predicted during the dry season due to the consistent decrease in rainfall during that period (see Figure 3). At the same time, CC is expected to cause an overall decrease in the total streamflow also due to a decrease in rainfall on the watershed scale (Table 6 and Figure 7).

**Table 6.** Seasonal percent change in the (WBCs) of the Heeia wetland due to climate change (CC) relative to the baseline.

| Scenario | Season | Rainfall | Streamflow | Runoff | Lateralflow | Baseflow | Recharge | Soil Moisture | ET | PET |
|---|---|---|---|---|---|---|---|---|---|---|
| Midmax 4.5 | wet | −3.47 | −10.53 | −7.41 | −5.15 | −18.94 | −12.11 | −7.83 | −1.86 | 2.71 |
| | dry | −22.52 | −19.37 | −33.11 | −29.15 | −14.87 | −64.44 | −20.94 | −16.68 | 5.98 |
| Midmax 8.5 | wet | −1.64 | −10.22 | −5.59 | −4.63 | −19.75 | −12.55 | −8.77 | 1.01 | 3.79 |
| | dry | −20.20 | −19.10 | −30.63 | −25.40 | −15.83 | −68.16 | −21.64 | −14.61 | 6.30 |
| Latemin 4.5 | wet | −4.77 | −14.84 | −11.20 | −8.19 | −25.12 | −18.45 | −10.74 | −0.09 | 3.79 |
| | dry | −18.22 | −22.47 | −28.70 | −23.97 | −21.21 | −66.95 | −21.65 | −14.10 | 6.30 |
| Latemin 8.5 | wet | −4.16 | −16.64 | −11.30 | −8.65 | −29.53 | −20.84 | −13.73 | 0.04 | 4.81 |
| | dry | −30.05 | −28.98 | −44.47 | −37.06 | −24.68 | −84.61 | −31.25 | −22.46 | 8.72 |

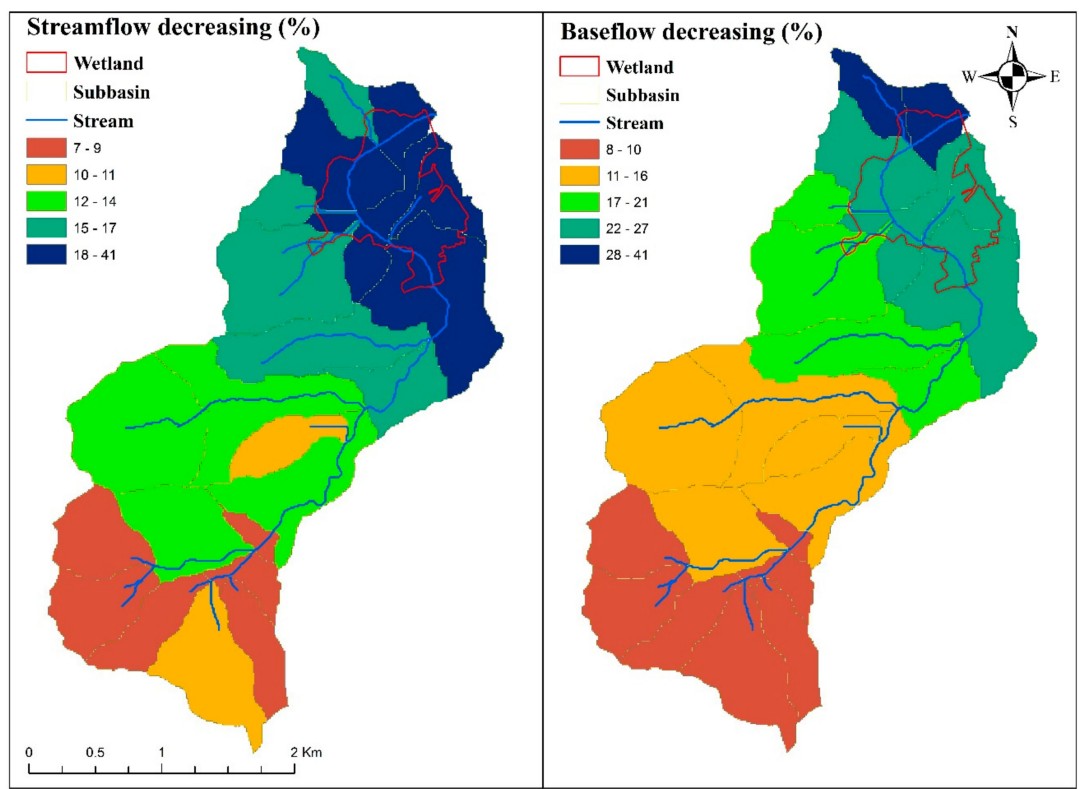

**Figure 7.** Average percent change in the (WBCs) of the Heeia watershed due to CC (Latemin 8.5 scenario) relative to the baseline.

## 3.4. Combined Effects of CC and LUC on WBCs

The results in Table 7 indicate that the total streamflow is expected to increase during the wet season, except under the minimum rainfall condition of RCP 8.5, indicating the positive effects of wetland restoration. However, streamflow is projected to consistently decrease during the dry season due to a more pronounced decrease in rainfall in that period. In addition, the combined LUC and CC scenario is expected to cause a larger negative change in baseflow (Table 7) because of the compacted soil layer below the soil surface of the taro field and an overall decrease in rainfall. LUC is predicted to cause an increased lateral flow, especially during the dry season, compared to the CC scenario (Tables 6 and 7). The soil moisture content is predicted to increase for LUC under all CC scenarios due to the ponding water in taro patches and to the created pool (Table 7). The trend of potential evapotranspiration is predicted to decline, especially during the dry season. However, the reasons behind the trend of the monthly actual evapotranspiration were not clear for all scenarios, due to the coupled interaction between the LUC and climate projections.

**Table 7.** Seasonally-averaged percent change in the WBCs of the Heeia wetland due to combined CC and LUC effect relative to the baseline.

| Scenario | Season | Rainfall | Streamflow | Runoff | Lateral Flow | Baseflow | Recharge | Soil Moisture | ET | PET |
|---|---|---|---|---|---|---|---|---|---|---|
| LUs2_Midmax 4.5 | wet | −4.15 | 4.60 | 57.72 | 28.03 | −54.19 | −50.57 | 13.54 | −4.80 | 2.30 |
| | dry | −14.63 | −28.89 | −16.64 | 51.68 | −51.82 | −73.63 | 32.22 | −4.01 | 4.15 |
| LUs2_Midmax 8.5 | wet | −4.15 | 4.60 | 57.72 | 28.03 | −54.19 | −50.57 | 13.54 | −4.80 | 2.30 |
| | dry | −14.63 | −28.89 | −16.64 | 51.68 | −51.82 | −73.63 | 32.22 | −4.01 | 4.15 |
| LUs2_Latemin 4.5 | wet | −4.77 | 1.38 | 52.54 | 25.16 | −56.75 | −52.52 | 11.00 | −3.92 | 3.22 |
| | dry | −18.22 | −32.73 | −24.11 | 43.72 | −54.06 | −80.57 | 24.98 | −7.08 | 6.54 |
| LUs2_Latemin 8.5 | wet | −4.16 | −1.12 | 50.21 | 22.49 | −59.13 | −53.70 | 7.32 | −3.06 | 4.14 |
| | dry | −30.05 | −39.78 | −41.95 | 22.98 | −56.00 | −91.59 | 13.95 | −15.31 | 8.97 |

max = maximum; min = minimum. ET = evapotranspiration; PET = potential evapotranspiration. LUs2 = wetland restoration for scenario 2 (decrease 50% of minimum flow).

The WBCs changes become more pronounced in the late 2080s period, especially during the dry season. The results in Table 7 indicated that the total streamflow was more correlated with changes in rainfall than other components, namely, temperature and solar radiation. In addition, the trend of the total streamflow was correlated with surface runoff, while the lateral flow was inversely proportional to surface runoff and total streamflow. The results also indicated that the CC effects were within the range of predictive uncertainties of the baseline. For instance, at the Haiku station, 95% prediction uncertainty (95PPU) of climate change scenarios was within the 95PPU of the baseline, which indicates no pronounced impacts of CC at the upstream subbasins (Figure 8). As a result, the effect of future climate change scenarios will be within the noise of model prediction uncertainty and may not have significant effect on the WBCs, except in the middle 2050s and late 2080s period of RCP 8.5 at the wetland scale. The findings indicated that the amount of streamflow is predicted to decrease particularly for the late 2080s of RCP 8.5 scenario (Figure 8).

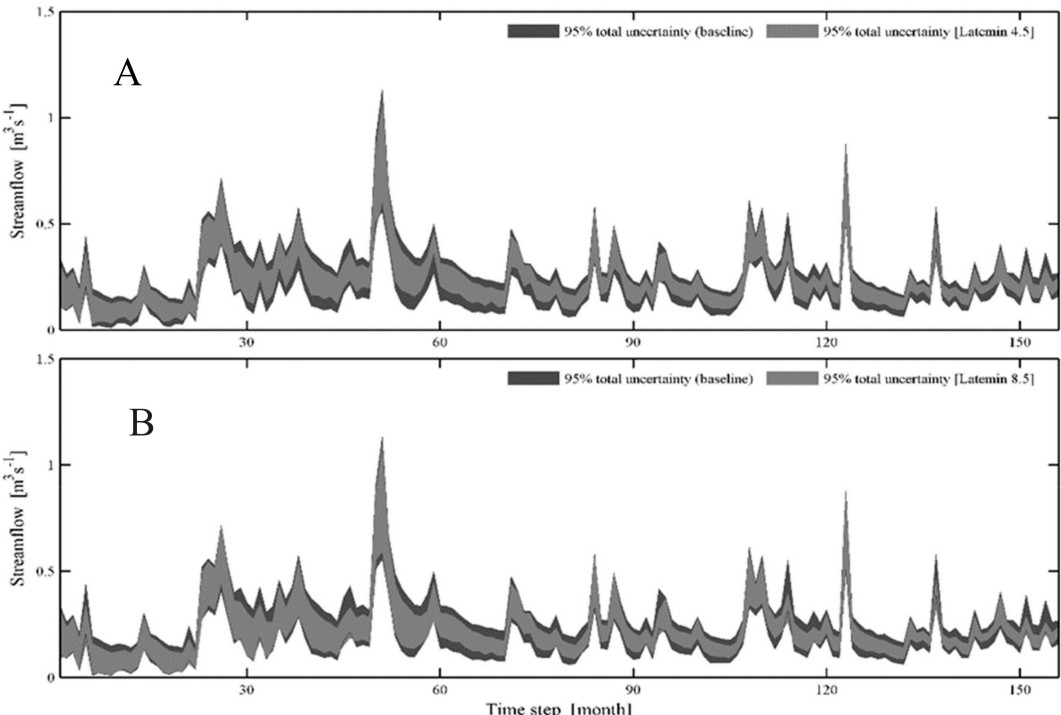

**Figure 8.** The monthly 95% streamflow prediction uncertainty for the thirteen years of baseline and as a result of rainfall, temperature, and solar radiation changes for (**A**) RCP 4.5 scenario and (**B**) 8.5 scenario at the wetland station.

## 4. Conclusions

This study assessed the effects of CC and LUC on the WBCs of the Heeia watershed using the SWAT model. The LUC scenario was consistent with the planned wetland restoration of the watershed. The LUC included the conversion of California grassland to taro fields and forested wetland (mangrove) to a retention pond. CC scenario used statistically downscaled rainfall data for Hawaii under RCP 4.5 and 8.5 scenarios while temperature and solar radiation changes were defined based on previous studies [19,43]. Baseline data of 2000 to 2014 were perturbed based on seasonal rainfall anomalies, temperature, and solar radiation changes.

Findings indicated that the combined CC and LUC scenarios will most likely cause an overall decline in WBCs during the dry season. On the other hand, recharge and baseflow were projected to decrease irrespective of the applied scenarios and seasons. Streamflow and surface runoff were expected to increase during the wet season under the combined LUC and CC scenarios. The spatial and temporal rainfall variations were the determinant factor of the relative negative impact on the

WBCs. Recharge and baseflow were highly sensitive to the combined effects of LUC and CC, especially during the dry season. Moreover, the down streamflow was significantly dependent on groundwater discharge during the dry season compared with other streamflow components. LUC had a larger effect than CC regarding the decrease in recharge, baseflow, soil moisture content, and streamflow, especially during the dry season. The WBCs were more affected in the late of the 2080s than 2050s periods. However, at the wetland scale, the CC effects were within the range of the uncertainties of both baseline and future climate change scenarios, except the middle 2050s and late 2080s period of RCP 8.5.

Overall, it is believed that the applied scenarios might negatively affect groundwater recharge and baseflow that in turn adversely influence the groundwater sustainability of the wetland and the ecological functioning of the coastal areas. Considering the importance of evaluating the hydrological changes after LUC, this study's results can provide useful information for evaluating future freshwater availability and designing appropriate mitigation measures to climate and land use changes. The study also assists in the decision-making process concerning implementing restoration efforts by community partners. Ultimately, such efforts will be instrumental towards a better management of land use and streamflows to achieve sustainable food production and healthy downstream coastal ecosystems.

**Author Contributions:** K.A.G. and O.T.L. conceived, designed, performed, analyzed, interpreted, and drafted the paper; A.I.E.-K. and H.D. conceived and supervised the research, contributed ideas during analysis, and edited the paper.

**Funding:** This paper was partly funded by a grant from the Pacific Regional Integrated Sciences and Assessments (Pacific RISA), NOAA Climate Program Office grant NA10OAR4310216 and by the National Oceanic and Atmospheric Administration, Project R/IR-19, which is sponsored by the University of Hawaii Sea Grant College Program, SOEST, under Institutional Grant No. NA14OAR4170071 from NOAA Office of Sea Grant, Department of Commerce. The views expressed herein are those of the authors and never reflect the funding agencies.

**Acknowledgments:** The authors thank the Kāko'o 'Ōiwi community for facilitating the research. Kariem A. Ghazal acknowledges the Iraqi ministry of Higher Education and Scientific Research for sponsoring his study at the University of Hawaii at Manoa. This is School of Ocean and Earth Science and Technology (SOEST) publication number 10699, UNIHI-SEAGRANT-JC-14-65, and a contributed paper WRRC-CP-2019-09 of the Water Resources Research Center (WRRC), University of Hawaii at Manoa, Honolulu, Hawaii.

**Conflicts of Interest:** The authors declare no conflict of interest.

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
