# Peer review of "Assessment of Wetland Restoration and Climate Change Impacts on Water Balance Components of the Heeia Coastal Wetland in Hawaii"

_hydrology, doi:10.3390/hydrology6020037_

Round 1

Reviewer 1 Report

This is an interesting case study investigating the combined impacts of wetland restoration and climate change on the water balance components of a coastal wetland in a Hawaiian island using a well-established hydrologic model SWAT. The intention of the manuscript is clear, and it is generally well-written and well-structured. The introduction section has rich information and the data and methods section also well-described. The discussion and conclusion sections could be elaborated to improve the quality of the manuscript.

General comments

1.       While the introduction section has rich information about the study area, the authors could include some more recent literature on the combined impacts of climate change and land use change on water balance in relatively lowland areas. See the following references.

Choi, W., Pan, F. and Wu, C., 2017. Impacts of climate change and urban growth on the streamflow of the Milwaukee River (Wisconsin, USA). Regional environmental change17(3), pp.889-899.

Psaris, M. and Chang, H., 2014. Assessing the impacts of climate change, urbanization, and filter strips on water quality using SWAT. International Journal of Geospatial and Environmental Research1(2), p.1.

2.       A summary of the results of sensitivity analysis, calibration and validation, including uncertainty and model performance evaluation would be very helpful for understanding the authors’ work. It should not be up to the readers to look at the other reference to verify the confidence of the authors’ analysis.

3.       The authors might want to discuss the potential caveats of using the approach described below.

…To capture the lower (minimum rainfall) and upper (maximum rainfall) anomalies per sub-basin, the sub-basins’ observed daily rainfall values were perturbed based on the estimated lower and upper rainfall anomalies (%)”

Considering an elevation gradient within the study area, the uncertainty of projecting future precipitation might be higher in some sub-basins than the other sub-basins where elevation gradient is low.

4.       A relatively a short period of 13 years of data were used as a baseline period. Given that most climate impact studies use a 30 year historical period as a baseline period, the authors should justify why a short period was used in their analysis.

5.       Since the authors have rich information at the sub-basin scale, why not reporting the combined impacts at the subbasin scale? Since some sub-basins contain a large area of wetland, the impacts might be more pronounced at a sub-basin scale. The authors could map changes in some water balance components under different scenarios.  

6.       The results and discussion section can be expanded more to include more detailed discussions of the results and their possible causes. There are lots of good results in Table 2, but the discussion is not robust enough to show why these findings are significant and how their results are comparable to other study results.

7.       The conclusions section could also better highlight why the projected changes are important and what can be done to alleviate the possible impacts.

8.       Grammars could be tightened as there are a few typos and missing punctuation marks. See my comments.

Some specific comments

Page 1, line 28. Can the authors quantify how much streamflow will decline?

Page 2, line 46. Spell out ET

Page 2, line 47. Insert comma before “which”

Page 2, line 60. “the local expected impacts of climate change includes” Check grammar.

Page 2, line 62. Insert “and” before “increase”

Page 2, line 64. “environmental warming”? “environmental” could be omitted.

Page 2, lines 86-87. Were there any previous studies or none in the windward side of the Oahu? Please clarify. If there were a limited number, please cite them.

Page 3, lines 111-113. Based on what year of the data?

Page 3, line 126. It is a little bit hard to follow the order of the four maps. Perhaps add arrows to navigate?

Page 3, line 162. “Based on previous studies” Please cite these previous studies.

Page 3, lines 169-171. Last sentence in the paragraph. It is probably better to place this sentence after describing wetland scenarios.

Page 3, line 181. “8.5km2” change “2” to superscript.

Page 7. Table 2 has a lot of information and hard to grasp at 1st glance. Better to separate into two tables. At least, the authors need to describe that the first 13 rows are for the results at the wetland scale.

Page 8. Below Table 2. “recharge and baseflow “are” projected to decrease…”  “Insert “are”

Page 8. “The results in Error! Reference source not found.” Please fix it.

Author Response

Ø  We thank the reviewer for providing detailed comments to improve the quality of the manuscript. Below are our responses to the comments.

English language and style

( ) Extensive editing of English language and style required 
( ) Moderate English changes required 
(x) English language and style are fine/minor spell check required 
( ) I don't feel qualified to judge about the English language and style 

Yes

Can be improved

Must be improved

Not applicable

Does the introduction provide sufficient background and   include all relevant references?

( )

(x)

( )

( )

Is the research design appropriate?

( )

(x)

( )

( )

Are the methods adequately described?

( )

( )

(x)

( )

Are the results clearly presented?

( )

( )

(x)

( )

Are the conclusions supported by the results?

( )

(x)

( )

( )

Comments and Suggestions for Authors

This is an interesting case study investigating the combined impacts of wetland restoration and climate change on the water balance components of a coastal wetland in a Hawaiian island using a well-established hydrologic model SWAT. The intention of the manuscript is clear, and it is generally well-written and well-structured. The introduction section has rich information and the data and methods section also well-described. The discussion and conclusion sections could be elaborated to improve the quality of the manuscript.

General comments

1.       While the introduction section has rich information about the study area, the authors could include some more recent literature on the combined impacts of climate change and land use change on water balance in relatively lowland areas. See the following references.

Choi, W., Pan, F. and Wu, C., 2017. Impacts of climate change and urban growth on the streamflow of the Milwaukee River (Wisconsin, USA). Regional environmental change17(3), pp.889-899.

Psaris, M. and Chang, H., 2014. Assessing the impacts of climate change, urbanization, and filter strips on water quality using SWAT. International Journal of Geospatial and Environmental Research1(2), p.1.

Ø  References included (see page 2, line 51– 55). 

2.       A summary of the results of sensitivity analysis, calibration and validation, including uncertainty and model performance evaluation would be very helpful for understanding the authors’ work. It should not be up to the readers to look at the other reference to verify the confidence of the authors’ analysis.

Ø   We agreed with this comment and made revision (see tables 1&2, figures 3&4).

3.       The authors might want to discuss the potential caveats of using the approach described below.

…To capture the lower (minimum rainfall) and upper (maximum rainfall) anomalies per sub-basin, the sub-basins’ observed daily rainfall values were perturbed based on the estimated lower and upper rainfall anomalies (%)”

Ø  Since SWAT does not use gridded based rainfall data, we summarized the spatially interpolated and projected seasonal rainfall anomalies by calculating the minimum and maximum anomalies per sub-basin. By using the estimated minimum and maximum rainfall anomalies per sub-basin, we captured the possible future seasonal rainfall anomalies. Then, we perturbed the current (baseline) rainfall data by using the minimum and maximum rainfall anomalies that resulted in two future rainfall scenarios (one for minimum anomaly and one for maximum anomaly). Use of such approach also provides the possible changes/impacts on water balance components (see pages 8 to 9, lines 215-229).

Considering an elevation gradient within the study area, the uncertainty of projecting future precipitation might be higher in some sub-basins than the other sub-basins where elevation gradient is low.

Ø  Since we used spatially interpolated rainfall anomalies per sub-basins and the future projections included the topographic effects, we believe that is handled in the projection.

4.       A relatively a short period of 13 years of data were used as a baseline period. Given that most climate impact studies use a 30 year historical period as a baseline period, the authors should justify why a short period was used in their analysis.

Ø  We certainly agree with this comment but due to lack of climate data. we opted to use the available data as much as possible but we recommend to use long-term data in the future[1].

5.       Since the authors have rich information at the sub-basin scale, why not reporting the combined impacts at the subbasin scale? Since some sub-basins contain a large area of wetland, the impacts might be more pronounced at a sub-basin scale. The authors could map changes in some water balance components under different scenarios. 

Ø  Clarified, please see pages 12 &13, figure 6, figure 7, and tables 4 to 6.

6.       The results and discussion section can be expanded more to include more detailed discussions of the results and their possible causes. There are lots of good results in Table 2, but the discussion is not robust enough to show why these findings are significant and how their results are comparable to other study results

Ø   Expanded, please see pages 12 to 16.

7.       The conclusions section could also better highlight why the projected changes are important and what can be done to alleviate the possible impacts.

Ø  Expanded, please see page 16.

8.       Grammars could be tightened as there are a few typos and missing punctuation marks. See my comments.

Some specific comments

Page 1, line 28. Can the authors quantify how much streamflow will decline?

Ø  Quantified, see page 1, line 31.

Page 2, line 46. Spell out ET

Ø  Spelled, see page 2, line 58.

Page 2, line 47. Insert comma before “which”

Ø  Inserted, see page 2, line 59.

Page 2, line 60. “the local expected impacts of climate change includes” Check grammar.

Ø  Revised, seepage2, line86.

Page 2, line 62. Insert “and” before “increase”

Ø  Inserted, see page 2, line 72.

Page 2, line 64. “environmental warming”? “environmental” could be omitted.

Ø  Revised, see page 2, line 73.

Page 2, lines 86-87. Were there any previous studies or none in the windward side of the Oahu? Please clarify. If there were a limited number, please cite them.

Ø  Cited, see page 3, line 97.

Page 3, lines 111-113. Based on what year of the data?

Ø  The data was before the 1950s, see page 4. Line 146.

Page 3, line 126. It is a little bit hard to follow the order of the four maps. Perhaps add arrows to navigate?

Ø  Done, see pages4&5, figures1 & 2.

Page 3, line 162. “Based on previous studies” Please cite these previous studies.

Ø  Cited, see page 9, line 232.

Page 3, lines 169-171. Last sentence in the paragraph. It is probably better to place this sentence after describing wetland scenarios.

Ø  Replaced, see page 11, line 271.

Page 3, line 181. “8.5km2” change “2” to superscript.

Ø  Done, see page 11, line 255.

Page 7. Table 2 has a lot of information and hard to grasp at 1st glance. Better to separate into two tables. At least, the authors need to describe that the first 13 rows are for the results at the wetland scale.

Ø  Separated, see pages13to 15, tables 4 t0 6 and, figures 6 &7.

Page 8. Below Table 2. “recharge and baseflow “are” projected to decrease…”  “Insert “are”

Ø  Revised, see page 15 line 328.

Page 8. “The results in Error! Reference source not found.” Please fix it.

Ø  Fixed, see page 15, line 356.

References

1.            Leta, O.T.; El-Kadi, A.I.; Dulai, H.; Ghazal, K.A. Assessment of climate change impacts on water balance components of heeia watershed in hawaii. Journal of Hydrology: Regional Studies 2016, 8, 182-197.

Reviewer 2 Report

Major grammatical and sentence structure issues. The authors need to use an English editing service. The handling editor should have screened this paper for English editing before sending out to review.

The authors need to use less absolute language, especially when referring to future environmental changes.

The authors need to provide details about the SWAT model calibration, validation, and sensitivity for this specific project.

The authors also need to specify what your land-use change treatments were. The final figures should be presented in a way that show the relative effects of land-use change and climate change along with the coupled effects of both drivers. The discussion also needs to include some idea of what the cost and how realistic it would be to restore wetlands to different levels.

Author Response

Ø  We thank the reviewer for providing detailed comments to improve the quality of the manuscript. Below are our responses to the comments.

English language and style

(x) Extensive editing of English language and style required 
( ) Moderate English changes required 
( ) English language and style are fine/minor spell check required 
( ) I don't feel qualified to judge about the English language and style 

Yes

Can be improved

Must be improved

Not applicable

Does the introduction provide sufficient background and   include all relevant references?

( )

(x)

( )

( )

Is the research design appropriate?

( )

( )

(x)

( )

Are the methods adequately described?

( )

( )

(x)

( )

Are the results clearly presented?

( )

(x)

( )

( )

Are the conclusions supported by the results?

( )

( )

(x)

( )

Comments and Suggestions for Authors

Major grammatical and sentence structure issues. The authors need to use an English editing service. The handling editor should have screened this paper for English editing before sending out to review.

The authors need to use less absolute language, especially when referring to future environmental changes.

Ø Revised, please See page 1, lines 19 and 36, page2, lines 58, 67, 74, 76, and 122, page.

The authors need to provide details about the SWAT model calibration, validation, and sensitivity for this specific project.

Ø Provided, please see pages 6 to 8.

The authors also need to specify what your land-use change treatments were. The final figures should be presented in a way that show the relative effects of land-use change and climate change along with the coupled effects of both drivers.

Ø Illustrated, please, see pages 12 and 14.

The discussion also needs to include some idea of what the cost and how realistic it would be to restore wetlands to different levels.

Ø  It is a hard question to answer about the cost of wetland restoration because of the cultural value of the place. A lot of work is being done through volunteer work. In addition, grants that allow restoration of culturally important places such as eliminating invasive species and setting up agricultural parcels. I think there is more to this story then just the monetary value and hence the cost cannot be evaluated directly.

Ø 

Volunteer work of Heeia wetland restoration

Round 2

Reviewer 1 Report

The authors adequatley addressed all the concerns raised by the reviewers. 

Author Response

Thanks for your comments

Reviewer 2 Report

Overall the manuscript needs a thorough English review. Please see specific comments in the attached pdf

The authors also should not reference an earlier version of the SWAT model from an unpublished source. Use this publication to document what is needed from your site-specific SWAT model. 

More details are needed about what variables were changed to simulated LUC and CC.

Author Response

Reviewer #2

Open Review

English language and style

(x) Extensive editing of English language and style required 
( ) Moderate English changes required 
( ) English language and style are fine/minor spell check required 
( ) I don't feel qualified to judge about the English language and style 

Yes

Can be improved

Must be improved

Not applicable

Does the introduction provide sufficient background and include   all relevant references?

(x)

( )

( )

( )

Is the research design appropriate?

( )

(x)

( )

( )

Are the methods adequately described?

( )

( )

(x)

( )

Are the results clearly presented?

( )

(x)

( )

( )

Are the conclusions supported by the results?

(x)

( )

( )

( )

Comments and Suggestions for Authors

Overall the manuscript needs a thorough English review. Please see specific comments in the attached pdf

Ø  We thank the reviewer for providing detailed comments to improve the quality of the manuscript. Below are our responses to the comments in attached pdf.

Ø  Please, we were edited the following points:

1.     See page 1, lines, 18, 20, 21, 23, 29, and 35.

2.     See page 2, lines 55, 57, 58, and 85.

3.     See page 3, lines 98, 101 t0 103, 115, 132, and 136.

4.     See page 6, lines 170 to 186, and 188.

5.     See page 8, line 240.

The authors also should not reference an earlier version of the SWAT model from an unpublished source. Use this publication to document what is needed from your site-specific SWAT model.

Ø  We were added more specific details about SWAT model. Please, see pages 6 and 9 to 12.

More details are needed about what variables were changed to simulated LUC and CC.

Ø  We mentioned to the changed variables of LUC and CC in page 8, lines 239 to 242, 251 to261, and page 6, lines 191 to 228.
